# Multidimensional diffusion processes in dynamic online networks

**David Easley**, **Eleonora Patacchini**, **Christopher Rojas** *

Department of Economics, Cornell University, Ithaca, NY, United States of America

☯ These authors contributed equally to this work.
* car332@cornell.edu

**Data Availability Statement:** The data is located on the public repository figshare, and can be accessed at https://figshare.com/articles/data_zip/11660352.

**Funding:** The authors received no specific funding for this work.

## Abstract

We develop a dynamic matched sample estimation algorithm to distinguish peer influence and homophily effects on item adoption decisions in dynamic networks, with numerous items diffusing simultaneously. We infer preferences using a machine learning algorithm applied to previous adoption decisions, and we match agents using those inferred preferences. We show that ignoring previous adoption decisions leads to significantly overestimating the role of peer influence in the diffusion of information, mistakenly confounding influence-based contagion with diffusion driven by common preferences. Our matching-on-preferences algorithm with machine learning reduces the relative effect of peer influence on item adoption decisions in this network significantly more than matching on earlier adoption decisions, as well other observable characteristics. We also show significant and intuitive heterogeneity in the relative effect of peer influence.

## Introduction

In recent years, social scientists have become increasingly interested in the datasets generated by people interacting in online networks. When studying these networks, an important issue is to estimate whether and how individuals' decisions are affected by the decisions of their peers, a concept known as peer influence. Determining the impact of peer influence on the tendency of two individuals to behave similarly is challenging primarily due to social selection. Social selection, otherwise known as sorting or homophily, is the tendency of individuals to form and maintain links with those who are already like them [1], i.e. who have similar preferences. Due to social selection, individuals who are linked are already more similar than those who are not linked, and that similarity could be self-reinforcing even without the link. Hence, when we view behaviors or information spreading between connected individuals, we need to account for similar preferences when trying to estimate the impact of peer influence on contagion.

In this paper, we present a statistical framework for estimating the effect of peer influence on item (aka product) adoption decisions in large dynamic online networks. Our paper builds on a small literature in economics and computer science which uses matched sampling to estimate causal peer influence effects on product adoption. Matched sampling is a non-parametric method to control for confounding factors and to overcome selection bias by comparing

**Competing interests:** The authors have declared that no competing interests exist.

treated and non-treated observations that have similar values of the matching covariates ([2]). The underlying assumption in this technique is that the treatment and the reaction to treatment should be independent conditional on the observable characteristics used in the matching process; i.e., that those characteristics are proxies for the drivers of social selection. There are two distinctive features of our method with respect to the existing studies ([3], [4]). [3] adapt propensity score matching for use in a dynamic online network to study the diffusion of a single item which is adopted by a large number of agents. In contrast to [3], our method can be used to study high-dimensional diffusion processes. Online social networks typically feature diffusion of a multitude of items at any given time, and an individual can adopt as many items as she would like. [4] study the effect of peer influence on the adoption of a large number of items, but they apply their method to a static online network, and they estimate peer influence for a single time-period. In our model, at each time period, each individual can be exposed to treatments on a large number of items (as many as the number of items which the agent has not adopted yet), and the effect of peer influence may vary depending on the item, the agent, her local network, and the time period. In order to deal with the individual- and time-specific nature of peer influence, we develop a novel dynamic, distance-based matching procedure.

The key novelty of our method is that we use a machine learning algorithm to predict agents' adoptions by inferring a latent factor model of their preference types ([5]). Previous research has used demographic as well as behavioral variables to match agents, but has not matched agents with similar past adoption behaviors. In their application to the yahoo IM network, the authors of [3] include a large number of demographic and behavioral characteristics for matching. Demographic variables are not always available for online data, and even when they are, these variables alone are unlikely to adequately control for homophily. This is important because to the extent that the effects of homophily are underestimated, the impact of peer influence is over-estimated. Our method adopts the fundamental insight from collaborative filtering that users with similar past adoption behaviors have similar preferences. In addition, utilizing past adoption behaviors takes into account the importance of past contagion in shaping the evolution of preference types. However, directly utilizing past adoption behaviors to match agents may be ineffective because of the large number of items and the sparsity of individual adoption. The authors of [4] overcome this problem by matching dyads (pairs of agents) based on having an approximately equal measure of past adoption similarity for each dyad. This approach requires both of the agents in a dyad to have enough past adoption behaviors in order to infer their preferences. Moreover, just because two agents $u$, $v$ are each equally similar to a third agent $w$ does not imply that $u$ and $v$ adopted the same types of items in the past. Our method will instead first embed past adoption behaviors into a low-dimensional preference type vector, and then match agents with similar predicted vectors. The advantage of our method is that it ensures we match agents who adopted similar items in the past.

Our method utilizes a machine learning algorithm which has not previously been used for causal inference, and so our paper also contributes to the growing literature on machine learning for causal inference. Machine learning has recently become a popular approach in applied econometrics to deal with and extract information from big datasets with many covariates. Discussions of machine learning for applied econometrics can be found in [6], [7], and [8]. For other applications of machine learning to casual inference using different algorithms especially tailored to the study of heterogeneous effects see [9], [10].

We illustrate our method using data from Github, the popular website for collaborative software development. On GitHub, users can create open-source software projects known as *repositories*, or repos for short, and push code to them. Other users can contribute to the repo (if given permission), or just use the repo for their own purposes. GitHub also has resources to help users find and share interesting repos. In particular, GitHub includes a directed social

network, similar in design to Twitter, in which users can *follow* other users to receive notifications about some of their activities on repos. GitHub users can also *star* a repo they find interesting, which is similar to liking something on Facebook, and will result in a notification on the *activity feeds* of their followers. GitHub is known as the "Facebook for programmers" and large tech companies want to reach communities of developers with their open-source software on GitHub ([11]). At the same time, rich longitudinal data concerning user activities on public repos and social network links can be accessed and analyzed. This makes GitHub a useful environment in which to study the diffusion of knowledge about software.

The *adoption* decision we focus on is whether or not to star a particular repo (item), and we will consider an agent to have adopted a repo when they star it. Although starring a repo does not necessarily imply usage of its contents, stars are an example of a common type of online social network interaction. Our results show that the effect of peer influence on item adoption decisions, relative to common preferences, is much smaller than it would appear to be if we do not control for past adoption behaviors. In particular, we demonstrate that using our machine learning algorithm leads to a significantly lower estimate of peer influence. We also show that the marginal influence of additional adopting peers is diminishing, and we find that the peer influence effect is strongest immediately after a peer adopts. We further examine other ways that the importance of peer influence varies with the types of agents and items. Our algorithm detects lower peer influence in contexts where past influence is likely to be important in shaping current adoption behaviors, such as for items which are similar to those an agent has adopted in the past. It also detects lower peer influence in contexts where it is easier to learn about the item without the link, such as for highly popular items. Taken as a whole, our empirical findings suggest exposure is the main pathway for peer influence on Github. That is, peer influence occurs because the links spread information that the agent would not otherwise have access to, as opposed to other possible reasons for peer influence in product adoption choices, such as information cascades ([12]) and (local) network externalities ([13]).

## Materials and methods

The *adoption* decision we focus on is whether or not to star a particular repo, and we will consider an agent to have adopted a repo when they star it. Let $u$ denote agents, $i$ denote items, and $t$ denote periods (months). We will refer to the agents whom $u$ follows as her *leaders*. Our primary definition of *treatment* is that one or more of agent $u$'s leaders have recently adopted an item $i$, which agent $u$ has not yet adopted. In this context, peer influence can be thought of as the direct causal effect of the treatment on the adoption outcome for the treated agent. In our data, peer influence would lead to connected agents adopting more similar items than non-connected agents.

Specifically, the causal quantity we seek to identify is the *relative risk of adoption*, $RR_t = p_t^{(1)}/p_t^{(0)}$, where $p_t^{(1)}$ is the probability of adopting an item during period $t$ for a treated agent, and $p_t^{(0)}$ is the probability of adopting an item during period $t$ when not treated, for an agent who could have been treated. We are interested in the effect of treatment on the treated, which is the (unobserved) effect of withholding a treatment that has in fact been implemented. Of course we do not actually observe the counterfactual quantity $p_t^{(0)}$ and it must be estimated from the data. Our definition of peer influence is the same as the definition used in [3], allowing us to compare our results to earlier findings.

Our estimation strategy is based on dynamic, distance-based matching. The propensity score $e(X_{u,i,t}) = \Pr(T_{u,i,t} = 1|X_{u,i,t})$ is the probability that agent $u$ is treated on item $i$ in period $t$, where treatment status is denoted by $T_{u,i,t}$ and $X_{u,i,t}$ are a set of time-varying covariates that describe the agent and item. Our identification strategy relies on the standard assumptions

underlying matching studies. First, we need to assume *conditional unconfoundedness*, which is the assumption that the potential adoption outcomes $(Y_{u,i,t}^{(1)}, Y_{u,i,t}^{(0)})$ are independent of treatment $T_{u,i,t}$, conditional on the covariates $X_{u,i,t}$. The second assumption is *overlap*, which requires that units have positive probability of being treated and non-treated, $e(X_{u,i,t}) \in (0, 1)$.

Conditional unconfoundedness in social network analysis is challenging primarily due to homophily or social selection. Homophily occurs when agents form links due to similarity in attributes. If the homophilous attribute is unobserved and positively correlated with the target variable (i.e. adoption), it gives rise to a form of selection bias in which connected agents appear to behave more similarly than non-connected agents, confounding the effect of peer influence. In our setting, preferences for items is a key attribute driving link formation and adoption outcomes. However, we do not observe the preferences of agents, and we will instead control for a set of proxy variables intended to predict preferences. Our matching covariates $X_{u,i,t}$ include several *baseline characteristics*, which are directly observable features that provide information about experience and overall activity level, but do not provide information about the types of items that the agent adopted in the past. Note that we list the baseline covariates used for matching in the appendix. In most of our empirical analysis, we also include in $X_{u,i,t}$ additional covariates which summarize the types of items the agent adopted in the past, to better proxy for preferences. For our preferred empirical method, machine learning is used to predict a low-dimensional representation of the types of items which agents adopted in the past.

Our empirical method is summarized at a high-level in Fig 1, which lists the three processes we will discuss in greater detail, and shows hypothetical snippets of our input and output data for the first period. The input data on the left shows three agents with user ids 1,2 and 3, and two items with repo ids 1 and 4. We assume that agent 1 follows agent 2, but agent 3 does not follow agent 2. Since agent 2 stars repos 1 and 4, agent 1 is treated on both of these repos, while we assume that agent 3 is not treated on either of these repos. The example output data on the right shows what would happen if we match agent 3 with agent 1. We would see that agent 1 adopts repo 1, but so does agent 3. Agent 1 also adopts repo 4, which agent 3 does not adopt. Hence our method would conclude that the treatment of agent 1 on repo 4 caused the adoption, while the treatment of agent 1 on repo 1 did not cause the adoption.

Our empirical method to go from the input data on the left to the output data on the right consists of three processes: i) Assemble treated agent-items, ii) Infer preferences, and iii) Nearest-neighbor matching. We will now explain each of these processes in greater detail. All

**Inputs**

Adoptions

| user id | repo id | star_datetime |
|---|---|---|
| 1 | 1 | 0.2 |
| 1 | 4 | 0.8 |
| 2 | 1 | -1 |
| 2 | 4 | 0.5 |
| 3 | 1 | 0.1 |
| ... | ... | ... |

Link Formation

| follower id | leader id | link_datetime |
|---|---|---|
| 1 | 2 | -0.5 |
| ... | ... | ... |

**Processes**

1) Assemble Treated Agent-Items

2) Infer Preferences: Basline, WRMF, Languages

3) Nearest-Neighbor Matching

**Output**

| follower id | alt id | repo id | follower star datetime | alt star datetime |
|---|---|---|---|---|
| 1 | 3 | 1 | 0.2 | 0.1 |
| 1 | 3 | 4 | 0.8 | x |
| ... | ... | ... | ... | ... |

**Fig 1. Process map of our empirical method.** The primary inputs to our method are the adoptions (stars) and link formtion (follows) data, which are formatted as shown on the left. Our empirical method can be broken down into the three processes listed in the middle. An example of our final output dataset is shown on the right. In the example shown, which is during the first period, a follower is treated on two items and adopts them both. A counterfactual agent (alt id) is matched with the follower, and adopts one of the items also adopted by the follower.

of the Python code to implement these processes can be accessed at github.com/christopher-rojas/PeerEffectsGitHub. Our input data can be accessed at figshare.com/articles/data_zip/11660352.

### Assemble treated agent-items

The first process is to assemble the treated agent-items, but to do so we must precisely define which leader adoptions are considered treatment, and when treatment is said to occur. First, since adoption is a one-time event, our treatments have a hazard structure in which agents are only treated on items that they have not adopted prior to the start of a period. If agent $u$ adopts the item during period $t$, then we measure treatment based only on her leaders who had adopted the item as of the time that $u$ adopted; i.e., excluding those who adopt during period $t$ and after agent $u$. By doing so, we alleviate concerns related to a possible simultaneity of adoption choices (e.g. reflection, [14]). Also, we fix the time of treatment for agent $u$ and item $i$ so that it is either the beginning of the month, or it is the most recent time at which a leader adopted the item, whichever comes later. Our definition of treatment includes items that a leader may have adopted prior to link formation, but the treatment time cannot begin prior to link formation. Lastly, we restrict the duration of peer influence by only defining adoption by a leader as treatment during period $t$ if it occurred since the beginning of period $t - 3$, and before the end of period $t$. In the full dataset, leader adoption happens up to 32 months before follower adoption, but about 54% of the diffusion takes place within the current and previous 3 months. We have tried varying the duration parameter and we find only small differences in the peer influence estimate for values greater than 3.

In most of our empirical analysis we will focus on a definition of treatment which includes any item adopted by at least one of an agent's leaders since the beginning of $t - 3$. However, we will also analyze the effect of treatment intensity by varying the definition of treatment based on how many leaders adopted and how recently they adopted.

To make all of these details more concrete, see Fig 2 for an example of an agent who is treated on and adopts an item during period 1. The agent has two leaders who adopt the item, called Leader 1 and Leader 2. Leader 1 adopts the item in period -2 (Oct., 2012), while leader 2 adopts the item in period 1 (Jan., 2013). Note that both leader adoptions could be included in

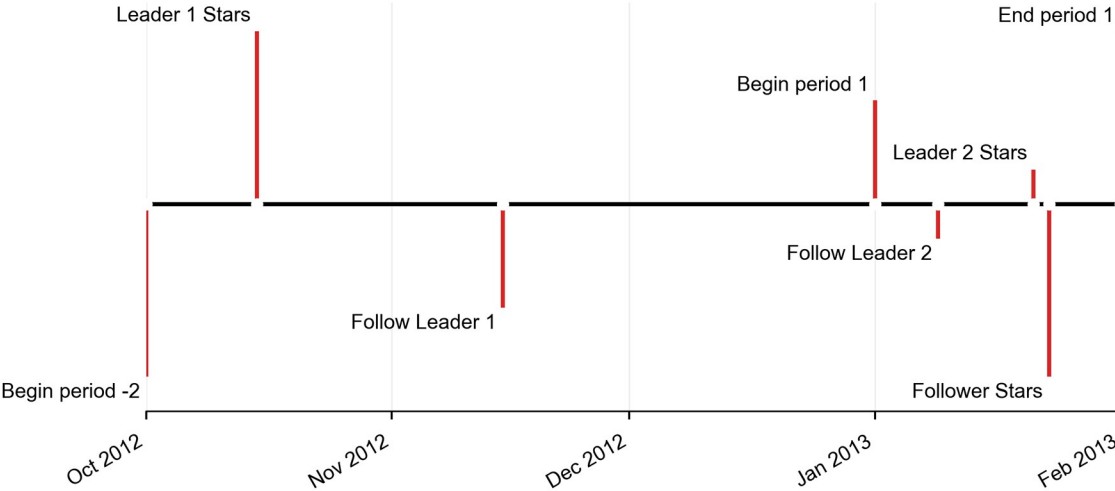

**Fig 2. Timeline of contagion.** An agent creates links to two other agents, known as Leader 1 and Leader 2. Leader 1 and Leader 2 each also adopt the item prior to the follower adopting the item.

the definition of treatment, because they both occurred since the beginning of period -2, and prior to the follower adopting. Hence, this agent would be treated regardless of whether we define treatment as having $\geq 1$ recent adopting leader, or having $\geq 2$ recent adopting leaders. However, regardless of the intensity of our treatment definition, the timing of the treatment will always be fixed at the moment that the more recent leader (Leader 2 in this example) adopts.

## Infer preferences

The next step is to estimate covariates which summarize past adoptions. There are many different ways in which we can add covariates that capture the types of items adopted in the past. We will consider several different methods, in order to demonstrate the superiority of our preferred method. The simplest way is to use the *adoption vector*, which is a long, sparse, binary vector which has a 1 for the *i*-th component if the agent *u* adopted the item *i* by the beginning of period *t*, and a 0 otherwise. However, the downside of this approach is that it would require matching on a huge number of covariates, and matching estimators are known to have greater finite-sample bias when the set of matching covariates becomes too large, relative to the number of observations ([15]). In our Results section we will show that matching directly on all past adoption choices leads to poor matching performance. Another simple approach would be to utilize an observable characteristic of the past adoptions which allows us to reduce the dimensionality of previous choices; in our setting, we use the programming languages of items adopted in the past. We define a language vector, which is a (much shorter) vector in which each component represents the fraction of items an agent adopted which are (primarily) written in a particular programming language. The downside of this approach is that our measure of past adoptions only measures one characteristic of the items, and there could be many other characteristics that are important to agents. In our Results section, we will also analyze the performance of past programming languages as a low-dimensional method to control for common preferences.

Our preferred estimation method will use a machine learning algorithm to infer latent agent and item characteristics which predict adoption choices. The *adoption matrix* is a matrix with a 1 in the (*u*, *i*) position if agent *u* has adopted item *i* before the beginning of period *t*, and 0 otherwise (i.e. it is the stacked matrix of adoption vectors). We use a well-known collaborative filtering algorithm for implicit feedback data (clicks, views, purchases, etc.), based on the weighted, regularized matrix factorization (WRMF) of the adoption matrix ([5]). The WRMF model is closely related to the singular value decomposition of the adoption matrix. We factorize the adoption matrix in a given period, to obtain a latent factor vector for each agent *u* and for each item *i*. Within the context of the model underlying WRMF, the latent factor vector for agent *u* represents *u*'s preferences over characteristics of items and the latent factor vector for item *i* represents the characteristics of item *i*. Agent *u*'s estimated payoff from any item is given by the inner-product of her latent factor vector with the item's latent factor vector. The latent factor vectors are chosen by the WRMF algorithm to best represent linear preferences over characteristics and the characteristics themselves, based on a static model of past adoption choices.

In the appendix we provide a detailed overview of the statistical model underlying WRMF. Note that the preferences learned by WRMF do not have a structural interpretation in our setting. The model underlying WRMF is static whereas our network is dynamic, and the WRMF model does not incorporate past contagion (which we address in a robustness check). However, we do not attempt to interpret the parameters estimated by WRMF in a meaningful way, we only use them as inputs for matching. What is important for our purpose is that the

preferences inferred by WRMF serve as effective proxy variables (in combination with the other baseline characteristics) for the true, unobserved preference types. We will provide evidence that this is indeed the case in our Results section.

Under the assumptions of the WRMF model, the weighted maximum-likelihood estimates of the preference representation will minimize the weighted (squared-Euclidean) distance between the adoption matrix and the preference representation, using a penalty function for the complexity of both the preference and characteristic representations. The posterior weighted log-likelihood equation for WRMF is derived in the appendix, and the estimation of the WRMF preferences is also detailed in the appendix.

The WRMF algorithm has 4 *hyper-parameters* (aka tuning parameters), which must also be estimated from the data. In order to set the hyper-parameters, we use the approach which is typical in the machine learning literature. We partition our data into a training set and a validation set; the training data is used to infer the model for a given value of the hyper-parameters, and the validation data is used to compute a statistic which measures the accuracy of the predicted preferences at out-of-sample prediction. We use random search over a set of possible hyper-parameter values, and select for our analysis the values of the hyper-parameters which optimize the value of the statistic computed on the validation data. Details are provided in the appendix.

## Nearest-neighbor matching

We propose a dynamic, distance-based matching strategy, with exact-matching on the item, and nearest-neighbor matching based on the distance between agent characteristics. Let $||\mathbf{X}||_\Sigma = \sqrt{\mathbf{X}'\Sigma\mathbf{X}}$ be the vector norm induced by the positive definite matrix $\Sigma$. We define $||\mathbf{X}_{u,i,t} - \mathbf{X}_{v,i,t}||_\Sigma$ as the distance between the vectors $\mathbf{X}_{u,i,t}$ and $\mathbf{X}_{v,i,t}$, where $\mathbf{X}_{v,i,t}$ represents the covariate values for a potential match for agent $u$ on item $i$. For the weighting matrix $\Sigma$ we use the (diagonal) inverse variance matrix. The inverse variance matrix is estimated once per period using all of the agents included in the data for the period. This weighting accounts for differences in the scale of the covariates and it places more weight on differences in covariates which do not vary much in the population than it does on ones which do vary a lot. Agents who have different covariates that others are very similar over are thus treated as significantly different; while agents who have different covariates that vary a lot in the population are treated as being more similar.

Note that in a typical application of distance-based matching to estimate the average treatment on the treated for a single treatment, one would include only the non-treated agents to estimate $\Sigma$ ([16]). We cannot do this because then we would need to estimate a separate $\Sigma$ for each item on which agents are treated. However, since the large majority of items have very few treated agents, it should be the case that for almost all items, $\Sigma$ for all agents is approximately equal to $\Sigma$ for only the non-treated agents for the particular item.

For each agent who is treated in each period, we compute their $M$ nearest neighbors, based on the past data. The past data we use is the preferences from above, as well as other agent features derived from past adoptions and link formations (we refer to these other, non-preference features as Baseline). $M$ is a parameter chosen by the analyst; we will have more to say about this once we discuss our matching algorithm in more detail below.

We finally have the pieces in place to match the nearest neighbor possible for each treated agent-item. Each period, we create a matched sample of treated and untreated agent-items. For each item $i$ on which an agent $u$ is treated during $t$, we select as $u$'s match the agent $v$ such that $u$ and $v$ are the nearest neighbors for $i$, and $v$ is a valid match for $u$ on $i$. In order for $v$ to be a valid match for $u$, we require that $v$ is not treated on $i$ during $t$, and we require that $v$ has

not adopted the item by the time of $u$'s treatment. We compute the nearest-neighbor for each treated observation (agent-item), with replacement, meaning that the same $v$ can be matched to $u$ for multiple items $i$. If we cannot find a valid non-treated agent in the first $M$ matches, then there is no match for agent $u$ and item $i$, and we remove this treated observation from consideration. $M$ needs to be large enough so that we are can match sufficiently many treated observations. We chose a value of $M = 50$ which allowed us to match over 99.99% of the observations. The benefit of our approach is that we only need to compute the nearest neighbors of each agent once per period, and so this approach can be used for the large number of items diffusing in our data.

## Measuring peer influence

Once we build our matched sample, for each item $i$ we can compute the fractions of treated $(\hat{p}_{i,t}^{(1)})$ and non-treated $(\hat{p}_{i,t}^{(0)})$ agents who adopt the item. The estimates for all of the items in the period are combined by weighting the estimate for each item by the fraction of exposed pairs for that item. Let $\alpha_{i,t}^{(1)}$ denote the fraction of treated observations in period $t$ which correspond to item $i$. We have

$$\hat{p}_t^{(1)} = \sum_i \alpha_{i,t}^{(1)} p_{i,t}^{(1)} \quad \hat{p}_t^{(0)} = \sum_i \alpha_{i,t}^{(1)} p_{i,t}^{(0)} \tag{1}$$

In our results, we report the ratio $\hat{p}_t^{(1)}/\hat{p}_t^{(0)}$. Note that $\hat{p}_t^{(1)}/\hat{p}_t^{(0)}$ simplifies to the number of treated agents who adopt an item, relative to the number of matched, non-treated agents who adopt an item. The probability of a follower adopting any given item is very small, but our statistic is uncontaminated by observations in which an agent does not adopt an item. In our Results section we will refer to the estimated treatment effect simply as $n_+/n_-$, ignoring the time period subscript.

Also, note that although we match pairs of agents, the statistic we use to measure peer influence relative to the effect of homophily uses only data about populations of matched pairs. That is, we do not directly compare matched agent-pairs; rather, we compare matched populations. We could instead match agents based on binning, and then directly compare binned populations, but this would require us to define bins of similar agents and items. Binning is problematic as the results may depend on the definitions of the bins and how they are filled. We view it as a benefit of our matching method that we do not need to specify bins.

Since our treated observations are not independent, we cannot rely on standard formulas to compute the asymptotic variance of our estimator. Instead, we will compute confidence intervals using bootstrap re-sampling simulations applied to the matched sample, similar to the inferential methodology used by [17] for the case of a single, binary treatment. We draw 100 samples from the matched pairs and compute the test statistic on each sample. We then use the 2.5 and 97.5 percentiles of the empirical distribution of the test statistic over the generated samples as the endpoints of our 95% confidence interval. A confidence interval with lower endpoint greater than 1 implies that the treatment has a significant effect.

## Data and descriptive evidence

Our data is time-stamped GitHub data collected from the GitHub Archive that includes user activities on public repos and social network links created over a 32-month period, from February, 2011 to October, 2013. We use this period of time because February, 2011 is the earliest time that GitHub Archive data is available, and October, 2013 is the latest date for which we observe the link creation time (GitHub stopped providing this data after October, 2013.). We

provide a more detailed background on GitHub in the appendix. We estimate peer influence during the 10 months from January to October, 2013; earlier data will be used for learning preferences, observing the evolution of the social network, and observing item adoptions. We define period 1 to be January, 2013; earlier months have period less than 1, and the final month is period $T = 10$. Since we will use earlier starring behaviors to learn preferences, we will restrict our data to include only agents who have enough stars to accurately infer their preferences. In order to do this, in a given period $t \geq 1$ we will estimate peer influence effects on the adoption behaviors of agents who have at least 10 stars prior to the start of the period. We also exclude inactive agents, which we define as agents that do not take any action for six consecutive months. Our final sample consists of 163,458 unique agents for whom we are able to estimate social influence in at least one month from January to October, 2013. These agents create a total of 10,036,987 stars of 855,317 unique repos, and they create a total of 1,662,393 links to 363,598 unique leaders. The agents in our sample are responsible for approximately 86% of the stars in our dataset, and they create 56% of the links. Summary statistics for our sample are provided in the appendix. On average, an agent stars 61 repos, and follows 10 leaders. A repo receives on average 12 stars, with a large dispersion about this mean value. Our analysis will also make use of the (primary) programming language of each repo, to provide a benchmark for how well we can match on preferences using only the features available in the data, without collaborative filtering. In the appendix, we graph the composition of stars by language over time. According to this measure, the most popular language is Javascript, followed by Ruby, and the popularity of the languages is fairly stable over time.

Fig 3(a) plots the fraction of adopters and non-adopters of a repo during period $T$, as a function of the number of leaders in their local network who have adopted the repo (since the beginning of period $T - 3$). The line with black dots is the (log-scale) probability of an adopter having $n$ adopting leaders, whereas the line with white dots is the probability for non-adopters. Fig 3(b) depicts the probability of adoption as a function of the number of leaders who have adopted the repo (since the beginning of $T - 3$). In Fig 3(a), we see that agents who star a repo during $T$ tend to follow significantly more agents who also starred that repo recently. In Fig 3(b), we see that the probability of adoption increases significantly when 1 or more agents whom an agent follows have starred the repo recently. Interestingly, the effect is non-linear. The probability of adoption decreases when more than 5 leaders whom an agent follows have adopted the repo, though it remains significantly higher than when 0 leaders have adopted.

In Fig 3(c) we implement a version of the well known "shuffle test" of social influence ([18]), modified for our setting. The original shuffle test is applied to a single item diffusing in an undirected social network. The test compares the clustering of adoption times between friends to the clustering between friends after permuting the adoption times such that the temporal frequency of adoptions remains constant. We implement the shuffle test in our setting by randomly permuting the adoption times for each item (so that the adoption frequency for the item over time remains constant), and we compare the distribution of dyadic (leader-follower pair) differences in adoption times for the original data relative to the data with the shuffled adoption times. We include only items with at least 10 total adoptions, which captures about 83% of all adoptions. Since our network is directed, we limit our comparison to observations in which the leader adopts before the follower. In comparison to the randomly permuted adoption times, we find that followers are about 260% more likely to adopt an item within the same day as the leader, and that the temporal interdependence persists; the follower is about 60% more likely to have adopted the item during the first 30 days with the actual adoption times.

Fig 3(a)–3(c) shows us that connected agents tend to adopt more similar items than non-connected agents, and that connected agents who adopt the same items do so closer together

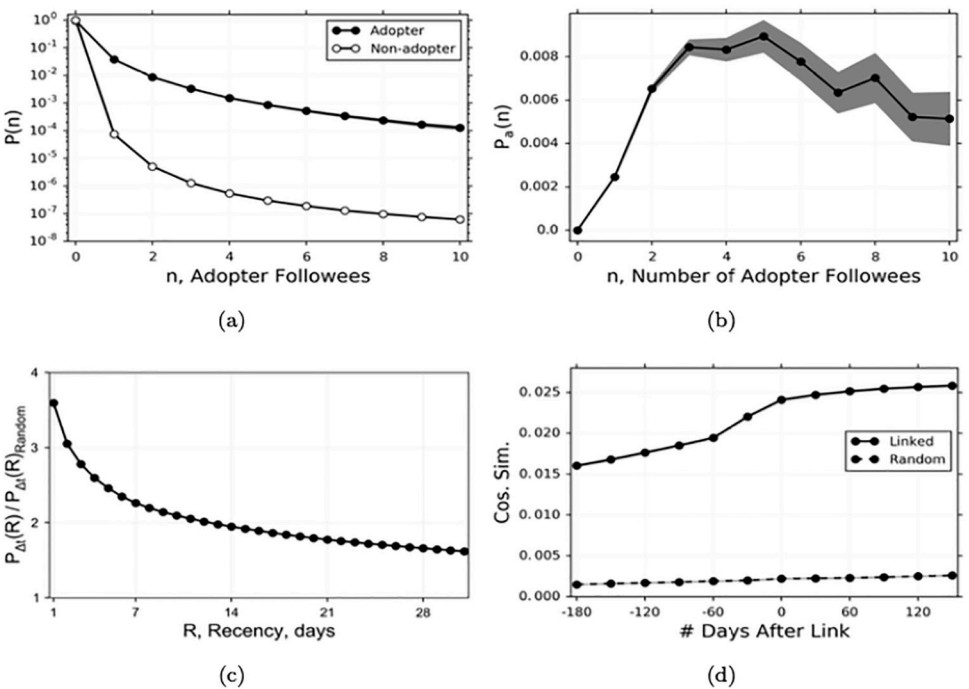

**Fig 3. Suggestive evidence of social influence and social selection.** (a) The fraction of adopters and non-adopters of a repo for whom $n$ of their leaders have adopted by the end of the period. (b) The probability an agent adopts a repo given $n$ of her leaders have adopted. The number of leaders who adopt is measured at the time the agent adopts. In (a) and (b), the results use data during period $T$. (c) The fraction of observed dyadic differences in adoption times between leader and follower with actual adoption times relative to randomly assigned adoption times. We include all adoptions for which the follower adopts an item after the leader. $\Delta t = t_i - t_j$, where $t_i$ is agent $i$'s (follower) adoption time, and $t_j$ is agent $j$'s (leader) adoption time. $R$ is the recency in days. (d) The average cosine similarity of pairs of agents who form a link during 30-day time periods measured before and after link formation. For each dyad, the time periods are centered so that period 0 is the period which begins at the moment of link formation. In (a)—(d), the gray shaded area indicates the 95% confidence interval.

in time than non-connected agents. Both of these findings are consistent with peer influence. However, this evidence is not sufficient to conclude that peer influence affects followers' adoption decisions. The choice of which repos to adopt, when to adopt and who to follow are endogenous. Homophily could cause connected agents to adopt similar items and to adopt them closer together. In Fig 3(d), we present evidence of homophily in our data. We compute the cosine similarity of the adoption vectors for pairs of agents who form a link and for random pairs of agents who do not form a link. For the agents who do form a link, we compute the similarity of their adoption vectors at 30-day intervals leading up to and after the moment of link formation. Note that cosine similarity between two non-zero vectors $v_1$, $v_2$ is defined as

$$\cos(v_1, v_2) = \frac{v_1 \cdot v_2}{||v_1||_2 ||v_2||_2}.$$

We include in our comparison all of the dyads for which the follower is one of the valid agents as defined above, and which we can observe for 180 days before and after the link forms. For the agents who do not form a link, we fix an arbitrary date of 5/1/2013, and compute the cosine similarity during 30 day intervals before and after this date. First, note that the adoption behaviors of pairs of agents who form links were already far more similar than those of pairs of agents who do not form links, even several months prior to link formation. This is consistent with social selection and could explain the patterns in Fig 3(a) and 3(b). Second, note that the

adoption behaviors of pairs of agents who form a link become more similar prior to link formation. This is consistent with increasing homophily during the time leading up to link formation. Even though the adoption behaviors continue to grow more similar after link formation, that additional growth may be caused by increasing homophily rather than influence. In order to determine the impact of peer influence on adoption decisions, we need to use our empirical methodology which allows us to account for the impact of attributes driving link formation and adoption choices.

## Results

### Baseline

We will now show that our preferred empirical method is successful at controlling for common preferences. First, we use simulations to analyze the finite-sample bias of our matching algorithm, under the assumption that homophilous adoption choices are determined by the WRMF preference types with the optimal hyper-parameter values. We simulate the adoption choices during the first period by assuming that each observed adoption is based on one of three possible mechanisms:

1. **Influence**: An agent adopts the item most-recently adopted by one of her leaders, which she has not already adopted.

2. **Homphily**: An agent adopts her most preferred but not yet adopted item, as predicted by WRMF.

3. **External Exposure**: An agent adopts the most popular item (based on adoptions since the beginning of period -2) which she has not already adopted.

In the simulations we randomly assign 90% of the adoptions during the first period to be determined by the homophily rule, 5% to be determined by the influence rule, and 5% to be determined by the external exposure rule (similar to [4]). We then replace the actual adoption choice in the data with the adoption choice generated by the randomly selected rule, while keeping the timing the same. We also estimate the nearest-neighbor for each agent, based on our preferred set of matching covariates which includes WRMF, prior to the beginning of period 1. We are interested in the ability of our matching method to estimate the counterfactual number of agents who adopt the item; i.e., the number of agents who adopt the item when not treated, for agents who would have been treated.

The results of our simulation exercise are shown in Table 1. In the results, the true counterfactual is given by the number of treated items adopted by the treated agent due to the homophily rule (10,406) plus the number of treated items adopted by the treated agent due to the external exposure rule (2,449). The estimated counterfactual is given by the number of matched, non-treated agents who adopt the item (10,907). Thus, we see that the estimated counterfactual performs fairly well; it generates approximately 84.8% as many adoptions as the

**Table 1. Simulation results for the number of items adopted by agents under the true counterfactual and the estimated (est.) counterfactual.**

| Adoption Type | Treated | # Adoptions | 95% C.I. |
|---|---|---|---|
| Homophily | Yes | 10, 406 | [10, 229.93, 10, 572.28] |
| External Exposure | Yes | 2, 449 | [2, 368.48, 2, 547.58] |
| **Est. Counterfactual** | **No** | **10, 907** | **[10, 707.95, 11, 064.1]** |

Est. Counterfactual refers to the matched, non-treated agents who adopt the item. The 95% C.I. is estimated using the non-parametric bootstrap.

true counterfactual. Table 1 also shows the confidence intervals for the number of adoptions from homophily and external exposure, as well as from the estimated counterfactual, based on bootstrap resampling simulations.

Of course, as we stated earlier, the true underlying preference types are different from WRMF and so simulations alone can only establish that our matching algorithm has low finite-sample bias when we observe preference types. We still need to establish that our matching covariates are effective proxies for the true unobserved preference types. We do this by providing evidence that our covariates used to match agents in the first period are related to adoption choices during the first period. Fig 4(a) provides an overview of the most-preferred items in period 1, as predicted by WRMF with the optimal hyper-parameter values, for each agent and each item, based on adoption behaviors prior to the beginning of period 1. Fig 4(a) shows that the preference types learned by WRMF are reasonable, in the sense that agents are more likely to adopt items for which they had a higher predicted preference. The correlation between predicted ranking and adoption probability is negative and highly significant ($-0.013$, $p < 0.001$). Fig 4(b) provides an overview of adoption similarity during period 1 for an agent and her nearest neighbors. In Fig 4(b), we use the adoption vector during period 1, which is a vector with a 1 in the $i$-th position if the agent adopts $i$ during period 1, and a 0 otherwise. We see that the cosine similarity of items adopted during period 1 is higher for pairs of agents who are lower-ranked neighbors than for pairs of agents who are higher-ranked neighbors. Once again the correlation between distance-ranking and adoption similarity is negative and highly significant ($-0.012$, $p < 0.001$). Fig 4(a) and 4(b) provide evidence that our covariates are effective proxies for the true adoption preferences during the first period. The low p-values for the correlation coefficients indicate a real relationship between similarity in terms of our matching covariates, and revealed adoption preferences.

Finally, to demonstrate that WRMF improves our ability to reduce homophily bias in the estimation of peer influence, we will compare several different matching algorithms. Homophily generally leads to an upward bias in the estimated peer influence effect, and our goal is to compare the ability of the different algorithms to attenuate this bias. We consider the following five algorithms:

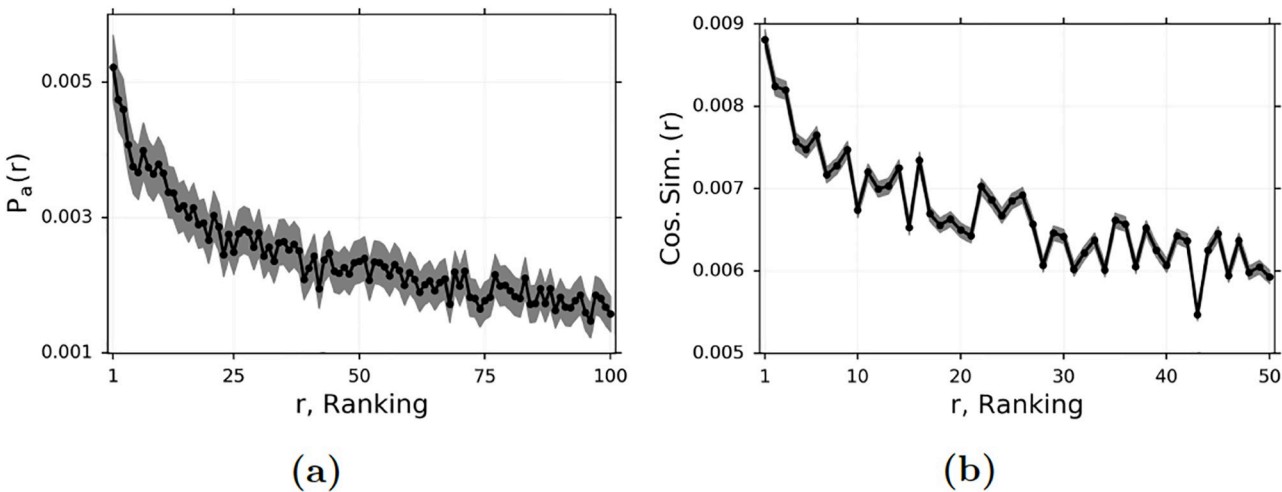

**Fig 4. Evidence that WRMF and other matching covariates are proxies for true, underlying preferences.** (a) The likelihood an agent adopts her $r$ ranked repo, as predicted by WRMF, during the period. (b) The cosine similarity of an agent's $r$-th nearest-neighbor, as measured by our distance function including WRMF and other matching covariates. The gray shaded areas indicate the 95% confidence interval.

- **Random matching**. The matching process selects random, non-treated agents as controls for each treated observation. Since links are not created randomly, the use of random matching cannot control for homophily. In order to reduce variance, for each treated observation we will use as our counterfactual the adoption choice averaged over all $M$ random neighbors (excluding only those neighbors who have already adopted or are themselves treated on the item).

- **Matching based on baseline agent's characteristics (Baseline)**. Nearest-neighbor matching using the covariates listed in the appendix. Importantly, the covariates do not measure the types of items adopted in the past by agents. We include the nearest-neighbor matching on baseline characteristics in order to understand the bias created by excluding past adoption behaviors.

- **Baseline+Adoptions** Nearest-neighbor matching using the baseline variables plus the adoption vectors for past adoptions. We include this approach to demonstrate that matching performs poorly when we do not reduce the dimensionality of the adoption vectors.

- **Baseline+Languages**. Nearest-neighbor matching using the baseline variables plus the programming languages of past adoptions. We measure agents' preferences by using the programming languages of their adopted items. We encode the top-100 most popular programming languages using a one-hot scheme, and measure the fraction of an agent's adopted items in each language. This approach is included to investigate whether our machine learning algorithm reduces homophily bias more than any directly observable features of the data. Note that the dimensionality of both of languages and WRMF factors will be fixed at 100.

- **Baseline+WRMF**. Nearest-neighbor matching using the baseline variables plus the (normalized) WRMF preference type vectors. Note that the magnitude of the WRMF preference type vectors is proportional to the number of items adopted by the agent, which is a covariate already included in our baseline variables. Therefore, we normalize the WRMF preference type vectors, so they have unit length.

We present the results for period $t = 1$ in Table 2, which show that the use of WRMF preferences results in the lowest estimate of peer influence, significantly outperforming the other candidate sets of matching variables. With random matching during the first period, the fraction of treated adopters is 201.84 times greater than the fraction of non-treated adopters. When we compare these results to our nearest-neighbor matching algorithm based on WRMF, we see that the estimated peer influence effect is dramatically reduced. During the first period the fraction of treated adopters is only 2.05 times greater than the fraction of non-treated adopters under matched sampling with WRMF. The random matching result suggests

**Table 2. The performance of different matching algorithms during the first period.**

| Matching Variables | $n_+/n_-$ | 95% C.I. |
|---|---|---|
| Random | 201.84 | [196.5, 207.33] |
| Baseline | 5.57 | [5.38, 5.75] |
| Activities+Baseline | 18.18 | [17.4528, 18.862] |
| Languages+Baseline | 2.91 | [2.85, 2.99] |
| **WRMF+Baseline** | **2.05** | **[2.01, 2.09]** |

Treatment is defined as following at least one agent who adopted the item, versus none. The outcome measure is the fraction of treated to untreated adopters ($n_+/n_-$). The 95% confidence interval is computed using the non-parametric bootstrap.

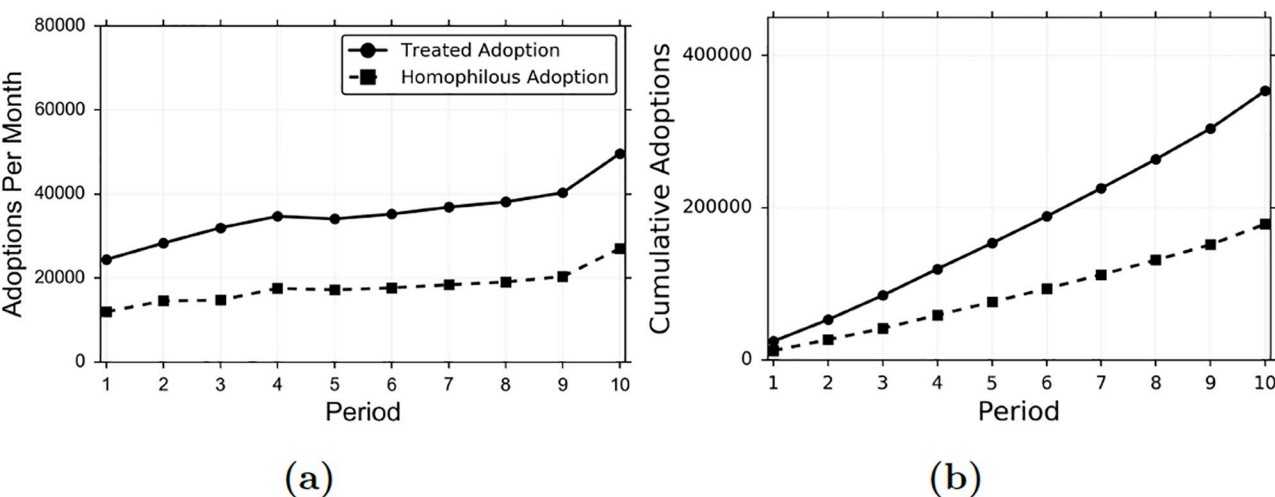

**Fig 5. Aggregate influence and homophily in multi-dimensional diffusion.** Treated and homophilous adoptions per month (a) and cumulatively over time (b). Homophilous adoption is adoption by a matched, non-treated agent, based on nearest-neighbor matching with WRMF.

that failing to account for selection into treatment results in an overestimation of the treatment effect by 9,744%. The language vectors result in the second lowest estimate, but it is still 42.42% higher than with WRMF. The inclusion of the adoption vectors actually causes the matching estimator to perform worse than the baseline matching approach which does not control at all for the types of items agent adopt in the past. Note that all of these results are similar in the other time periods as well.

In Fig 5 we depict the total amount of contagion for treated agents, and the total amount of contagion which can be attributed to homophily; i.e., the total amount of adoption of items by (WRMF) matched, non-treated agents, versus the total amount of adoption by treated agents. It appears that homophily can explain over half of the contagion that we see in the data (50.47%), which is similar to what was found in [3]. The estimated homophily without collaborative filtering would be much lower (0.8% with Random, 5.8% with Baseline +Adoption Vectors, 20.69% with Baseline, 32.30% with Baseline+Languages), thus producing an overestimation of the peer influence effect on contagion.

## Treatment intensity and timing

Next we turn our focus to different intensities of influence, based on the number of adopting leaders. Fig 6(a) and 6(b) shows a comparison of WRMF-based matching to random matching over time and for different numbers of adopting leaders. With random matching, the estimated treatment effect of having 2 and 3 (or more) adopting leaders is higher than that of having a single adopting leader (Fig 6a), and the results exhibit a downward time-trend. Matched sampling with WRMF results in peer influence estimates that are significantly lower, and the downward time-trend disappears (Fig 6b). The marginal influence of adopting leaders is actually diminishing, and the treatment level of 3 (or more) adopting leaders does not have a statistically significant peer influence effect in most of the time periods. One reason why random matching incorrectly suggests that marginal influence is increasing may be that it fails to control for the greater homophily (measured by cosine similarity of covariates) that exists with larger groups of adopting leaders, as shown in the inset in Fig 6b). See the appendix for the full list of attributes used to measure homophily.

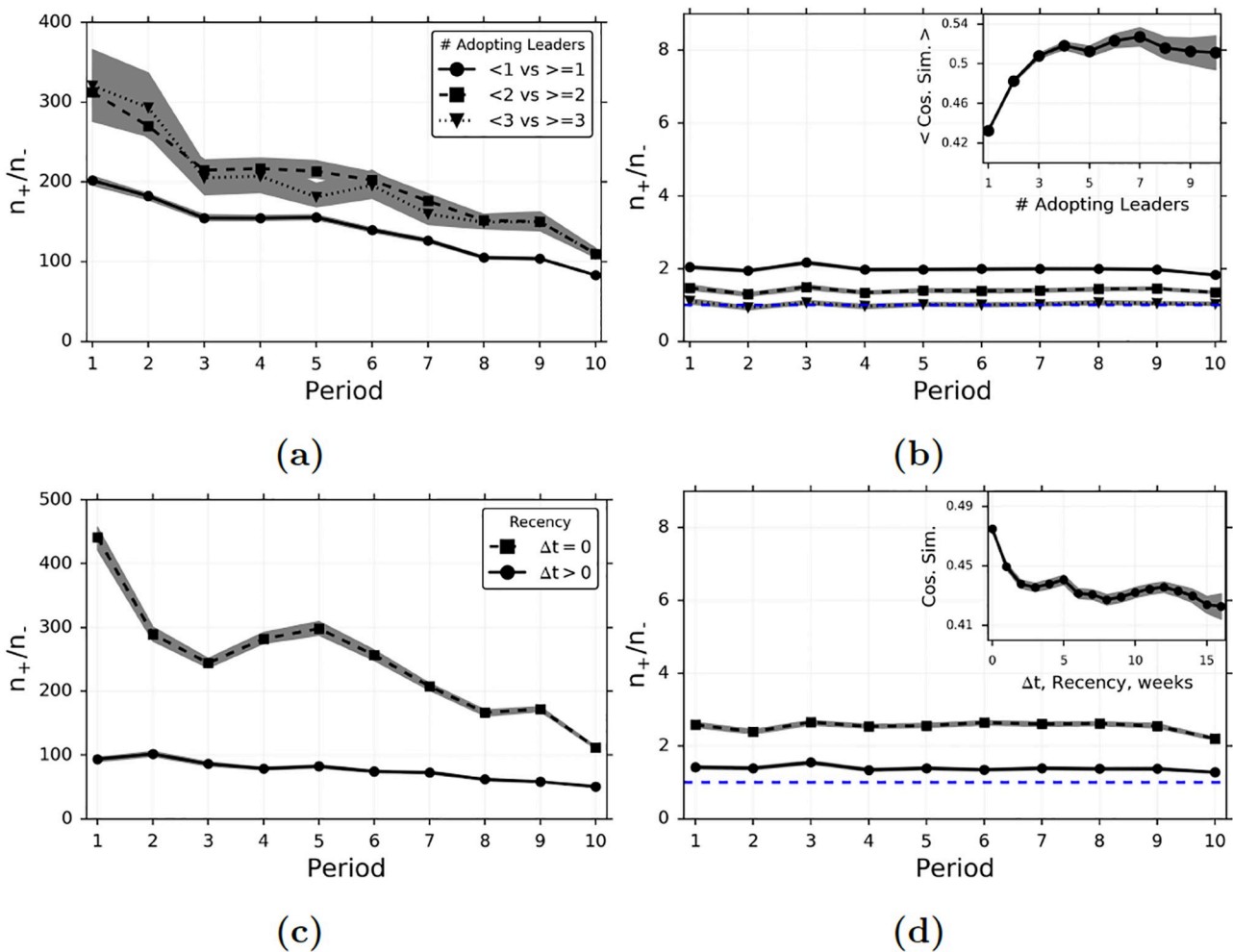

**Fig 6. Peer influence and homophily for different intensities of the treatment.** (a)-(b) The fraction of observed treated to untreated adopters ($n_+/n_-$) over time under random matching (a) and nearest-neighbor matching with WRMF (b), for different treatments defined by the number of adopting leader(s). The *inset in (b)* graphs the average cosine similarity between the attributes of an adopter and her adopting leaders, for different numbers of adopting leaders. We only include an observation if we can estimate preferences with WRMF for at least 75% of a follower's adopting leaders. (c)-(d) The fraction of observed treated to untreated adopters ($n_+/n_-$) over time under random matching (c) and nearest-neighbor matching with WRMF (d), for different treatments defined by the recency of leader adoption. In (c) and (d), $\Delta t$ refers to the number of periods between the current period and the most recent leader adoption. The *inset in (d)* graphs the dyadic cosine similarity between an adopting follower and her adopting leader, based on the amount of time between adoptions. In (b) and (d), the dashed blue line indicates a ratio of 1, which means the treatment has no effect. In (a)-(d), the 95% confidence intervals are computed using the non-parametric bootstrap. In the *insets in (b) and (d)*, the 95% confidence intervals are based on the formula for standard error.

We also examine how peer influence varies based on the timing of adoption decisions, by examining the effect of treatment based on time elapsed since an agent's leader adopted an item. Let $\Delta t_{u,i}$ denote the number of periods between the current period ($t$), and the most recent period that one of agent $u$'s leaders adopted the item. Fig 6(c) and 6(d) shows a comparison of random matching to WRMF-based matching over time and for items which an agent's leader(s) most-recently adopted during the current period ($\Delta t_{u,i} = 0$), or during an earlier period ($0 < \Delta t_{u,i} \leq 3$). With both random matching (Fig 6c) and WRMF-based matching (Fig 6d), the estimated treatment effect is larger when $\Delta t_{u,i} = 0$, and smaller when $\Delta t_{u,i} < 0$. In addition, random matching overestimates peer influence by more when $\Delta t_{u,i} = 0$ (17,000% in period 1) than when $\Delta t_{u,i} < 0$ (6,500% in period 1). This may be because there is greater

homophily between agents who adopt closer together in time (inset in Fig 6d). Overall, the results show that peer influence is strongest immediately after a leader adopts an item, which makes sense since an item is highest on a follower's activity feed immediately after the leader adopts. However, we also find that homophily plays a relatively greater role in driving contemporaneous adoptions. Hence, it is most important to control for homophily in precisely the setting when influence is also strongest.

## Treatment heterogeneity

We also use nearest-neighbor matching with WRMF to evaluate the treatment effect (of at least 1 adopting leader versus none) under various types of heterogeneity. To do this, we compare the fractions of treated and non-treated adopters for observations with different types of followers, leaders and/or items.

In Fig 7(a) we examine whether or not peer influence plays a greater role in adoption of items for agents who have adopted similar items in the past. We estimate similarity by using the cosine similarity between WRMF preference and characteristic type vectors. We group observations into those for which a treated agent had similarity above or below the average similarity for her treated items in the period (so we only include agents treated on at least two items). In Fig 7(b) we analyze heterogeneity by agents' levels of experience, measured using the total number of adopted items. We compare observations in which the follower has adopted at most 50 items versus observations in which the follower has adopted at least 75 items, measured as of the beginning of each period.

The results in Fig 7(a) and 7(b) are consistent with our algorithm successfully controlling for common preferences in instances when past influence may affect the evolution of preferences. In Fig 7(a) we see that adoption of items which are more similar to those an agent has adopted in the past tends to be driven less by peer influence (although the difference is not statistically significant in periods 5, 9 and 10). This result is in fact the opposite of what was found in [3], in which it was found that agents with a greater interest in news content are more susceptible to peer influence on an item that provides news content. In Fig 7(b) we see that agents who have adopted fewer items overall are more susceptible to influence than agents who have adopted more items. The difference between our methodology and the one adopted by [3] is that our algorithm takes into account past peer influence in shaping the evolution of preferences. Agents with high preference similarity to an item, as measured by WRMF, may have leaders who adopted more similar items in the past, so that past influence already caused them to adopt these types of items before. And agents who have adopted more items overall are also more likely to have been already exposed to peer influence on similar items, which affects the evolution of their preferences.

Previous research ([3]) suggests that homophily is greater amongst early adopters of an item, but has not found that peer influence varies during an item's life cycle. We investigate how peer influence and homophily vary for newer versus older items in our data. To do so, we define an item's age in a given period as the number of days since the item first appeared in the data, computed at the end of the period. We compare items which first appeared at most 180 days ago against items which first appeared at least 360 days ago. We do indeed find that adoption of newer items is driven significantly less by peer influence (Fig 7c), and the results are significant in every period, including period 6. We also find that greater homophily exists amongst the early adopters of an item (inset in Fig 7c).

We have already seen that peer influence plays a smaller role in the adoption of items more similar to an agent's preference type, which is consistent with exposure as the primary reason for peer influence. On GitHub there is a huge number of items diffusing simultaneously,

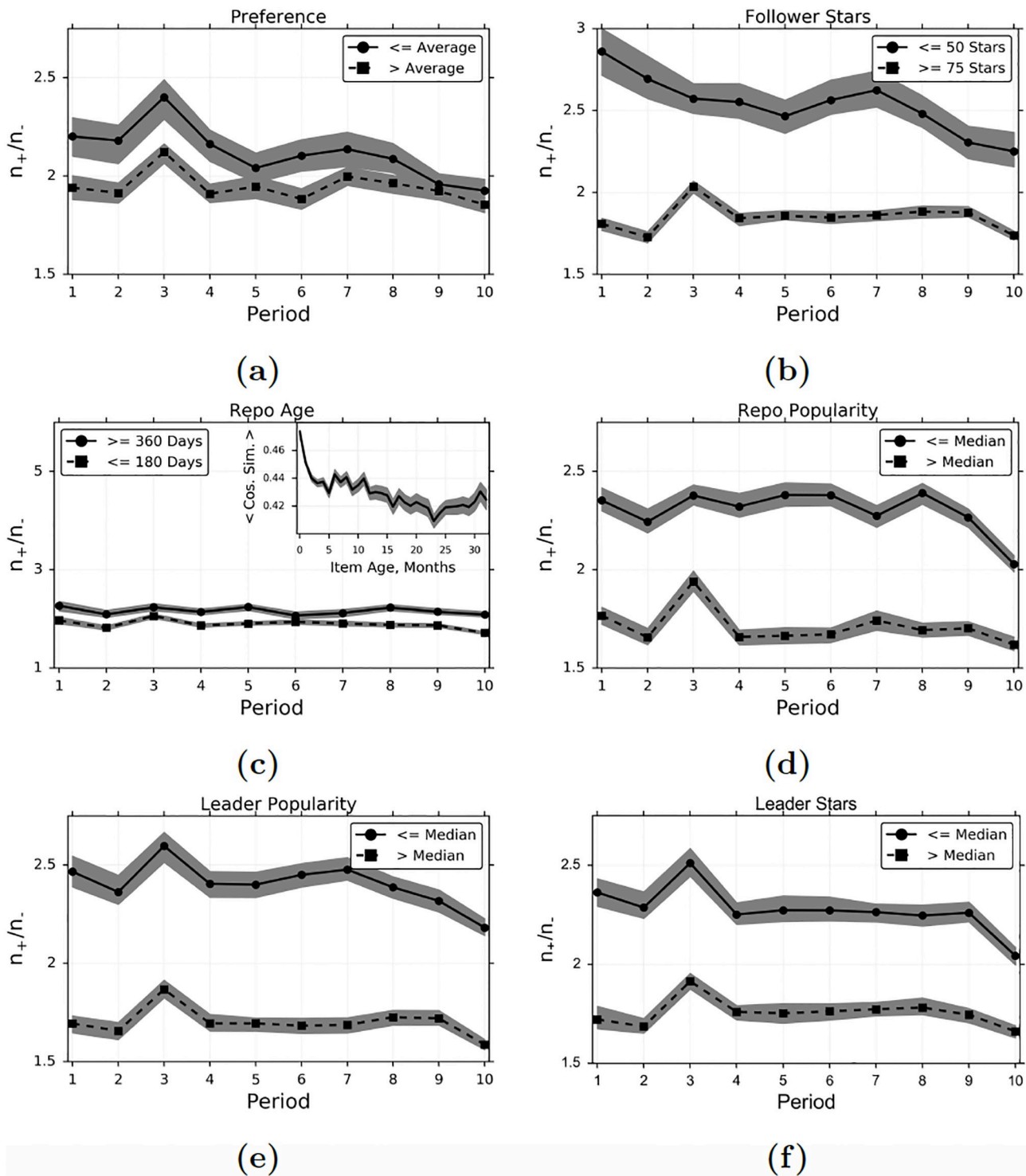

**Fig 7. Heterogeneity in peer influence and homophily.** (a) An agent's preference for a repo is above or below her average preference for treated repos during the period. We only include agents who are treated in a period on at least 2 repos. (b) The number of stars by the follower up to the end of the period $t - 1$ is at most 50, or at least 75. (c) The repo first appeared at most 180 days ago, or at least 360 days ago, measured at the end of the period. The *inset in (c)* graphs the average similarity of an adopter to each of her adopting leaders, based on the number of months since the item first appeared in the data. (d) The number of stars of the repo, as of the end of period $t - 1$, is above or below the median for all observations in the period. (e) The average (log+1) number of followers of the adopting leaders, as of the end of period $t - 1$, is above or below the median for all the observations in the period. (f) The average (log+1) number of stars by all of an agent's leaders, as of the end of period $t - 1$, is above or below the median for all observations in the period. In (a)-(f), treatment is defined as having at least one leader who adopts the repo, and the 95% confidence interval is computed using the non-parametric bootstrap. In the *inset in (c)*, the 95% confidence is based on the formula for standard error.

making it difficult for an agent to discover an item less similar to her previously adopted items, without contagion. We look for additional evidence for exposure by examining heterogeneity with respect to popularity, as well as the number of items adopted by an agent's leaders. As mentioned earlier, popular items may be easier for agents with similar interests to learn about, because more popular items have avenues other than contagion through which agents can learn about them. For example, GitHub (similar to other online platforms) has a special page listing recently popular repos, which are known as trending repos. Popular repos may also be easier to find outside of GitHub, on social media and independent websites. We measure the popularity of an item in period *t* based on whether or not the item has (log+1) number of adopters above or below the median for all observations during the period. We find that peer influence is significantly lower for more popular items than for less popular items (Fig 7d). It may also be easier for agents to learn about the items adopted by popular agents, even without following them, because of the availability of alternative channels. In addition to listing trending repos on a special page, GitHub also highlights trending programmers, defined as those agents who receive many new followers. Those agents may be well-known on other platforms popular with programmers too, such as Twitter or Stack Overflow. Previous research into Twitter has shown that popular agents are not necessarily more influential in terms of spawning retweets or mentions ([19]), but this research does not estimate causal peer influence effects, which we now do. For each observation in a period, we define popular leaders to be those whose average number of followers is above the median for all observations in the period. We find that peer influence is significantly lower for items adopted by more popular leaders than for items adopted by less popular leaders (Fig 7e).

In Fig 7(f) we compare the behavior of followers in each period for whom the average (log+1) number of adoptions by their leaders, as of the end of the previous period, is either above or below the median for all observations that period. The results show that agents who follows leaders that adopt many items are less influencible than agents who follow leaders that adopt fewer items. On GitHub, followers get a notification for each star by one of their leaders. When an agent follows leaders who are creating a large number of stars, it seems reasonable to hypothesize that she is proportionally less likely to pay attention to each star created by a leader, because the higher volume of information being transmitted to the follower makes it more costly for her to find each individual item. Previous research has demonstrated in lab experiments that as the size of an individual's peer group increases, each peer has less of an impact ([20]).

We believe that our findings with respect to the popularity of items and leaders, as well as the number of items adopted by an agent's leader(s), support the hypothesis that exposure is the main pathway underlying peer influence. It seems highly plausible that for each of these types of heterogeneity, the heterogeneous covariate affects the observed adoption outcomes primarily through its effect on whether an agent is aware of the item at all. Hence, the main reason that we observe peer influence is because a leader's adoption of an item increases the likelihood that the follower is exposed to an item to which she would not otherwise have been exposed. For items to which an agent was already likely to be exposed, even without leader adoption, the peer influence effect is smaller.

## Robustness checks

Finally, we provide a couple of robustness checks, which are summarized in greater detail in the appendix. First, we implement an extension of the WRMF algorithm which incorporates past exposure by placing greater weight on items which have been adopted by a leader in the estimation of the follower's preferences. The results for the first period are in Table 3, which

**Table 3. First period results with WRMF that does (Exposure = Yes) or does not (Exposure = No) take into account leader adoption.**

| Exposure | $n_+/n_-$ | 95% C.I. |
|---|---|---|
| **No** | **2.05** | **[2.01, 2.08]** |
| Yes | 2.17 | [2.12, 2.22] |

**Table 4. First period results with the 95% confidence intervals computed using an alternative bootstrap technique which accounts for two-way random effects.**

| Matching Variables | $n_+/n_-$ | 95% C.I. |
|---|---|---|
| Random | 201.84 | [181.81, 224.7] |
| Baseline | 5.57 | [5.05, 6.24] |
| Languages+Baseline | 2.91 | [2.69, 3.19] |
| **WRMF+Baseline** | **2.05** | **[1.92, 2.24]** |

shows that the extension reduces homophily bias less than our main specification. Second, we consider an alternative bootstrap technique which accounts for two-way random effects. The results for the first period are in Table 4, which shows that the alternative bootstrap technique does not change our main result.

## Concluding remarks

We present a novel statistical framework to estimate peer influence effects for a multidimensional diffusion process that uses machine learning to control for common preferences that drive adoption decisions. Using this technique allows us to account more effectively for the drivers of social selection. We show that ignoring common preferences leads to significantly overestimating the role of peer influence in the diffusion of knowledge on GitHub, mistakenly identifying homophily-driven diffusion as influence-based contagion. In particular, we show that our preference-based matching algorithm is able to reduce the estimate of peer influence effects significantly more than matching with other observable variables in the data. We also find significant heterogeneity in peer influence for different types of items, leaders, and followers. Peer influence is lower for agents who have been influenced on similar items in the past, and it is lower for agents who are more likely to be exposed to an item even in the absence of peer adoption. Our findings thus point to the importance of exposure for diffusion processes in online networks.

Two caveats surrounding the interpretation of the results need to be highlighted. First, as it is common in the related literature ([3]), we assume that the social network is independent from the adoption behavior. Our peer influence effect ignores any indirect effects which may occur because of a change in the matching covariates as a result of the change in treatment status leading to a different equilibrium in which the matching covariates are different. On the other hand, a more exhaustive game-theoretic model of the problem would require a complex dynamic model of product adoption choices and social network formation, and the results would be conditional on the assumptions. Second, in our empirical context, we measure item adoption using starring of repos, which does not imply item usage. The application of our methodology to other social networks and adoption behaviors may generate different results.

Although we cannot be certain that we have captured all of the intricacies driving selection bias, our approach provides a significant improvement in the ability to sort out homophily and influence in a social network using observational data.

## Supporting information

**S1 File. Appendix with additional information.**
(PDF)

## Author Contributions

**Conceptualization:** David Easley, Eleonora Patacchini, Christopher Rojas.

**Data curation:** Christopher Rojas.

**Formal analysis:** David Easley, Eleonora Patacchini, Christopher Rojas.

**Investigation:** David Easley, Eleonora Patacchini, Christopher Rojas.

**Methodology:** David Easley, Eleonora Patacchini, Christopher Rojas.

**Software:** Christopher Rojas.

**Writing – original draft:** David Easley, Eleonora Patacchini, Christopher Rojas.

**Writing – review & editing:** David Easley, Eleonora Patacchini, Christopher Rojas.

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
