## [Decision Letter · Decision Letter 0]

9 Oct 2019

PONE-D-19-23200

Multidimensional diffusion processes in dynamic online networks

PLOS ONE

Dear Professor Easley,

Thank you for submitting your manuscript to PLOS ONE. After careful consideration, we feel that it has merit but does not fully meet PLOS ONE’s publication criteria as it currently stands. Therefore, we invite you to submit a revised version of the manuscript that addresses the points raised during the review process.

A number of issues have been raised by the two referees which need to be carefully addressed. In particular one of the referee ask to have access to the replication code. Please allow the referee to double check it.

We would appreciate receiving your revised manuscript by Nov 23 2019 11:59PM. To enhance the reproducibility of your results, we recommend that if applicable you deposit your laboratory protocols in protocols.io, where a protocol can be assigned its own identifier (DOI) such that it can be cited independently in the future. For instructions see: http://journals.plos.org/plosone/s/submission-guidelines#loc-laboratory-protocols

We look forward to receiving your revised manuscript.

Kind regards,

Massimo Riccaboni

Academic Editor

PLOS ONE

Journal Requirements:

1. We note that you have indicated that data from this study are available upon request. PLOS only allows data to be available upon request if there are legal or ethical restrictions on sharing data publicly. For more information on unacceptable data access restrictions, please see http://journals.plos.org/plosone/s/data-availability#loc-unacceptable-data-access-restrictions.

Reviewers' comments:

Reviewer's Responses to Questions

**Comments to the Author**

1. Is the manuscript technically sound, and do the data support the conclusions?

Reviewer #1: Yes

Reviewer #2: Yes

2. Has the statistical analysis been performed appropriately and rigorously? 

Reviewer #1: Yes

Reviewer #2: Yes

3. Have the authors made all data underlying the findings in their manuscript fully available?

Reviewer #1: Yes

Reviewer #2: Yes

4. Is the manuscript presented in an intelligible fashion and written in standard English?

Reviewer #1: Yes

Reviewer #2: Yes

5. Review Comments to the Author

Reviewer #1: Thank you for giving me the opportunity to read this interesting paper.

## What I really liked

I really like the matching procedure and the robustness checks, this must have been an enormous amount of work.

I especially liked the usage of collaborative filtering in conjunction with a regime of robustness checks, well done.

Given the complexity of the analysis and the numerous steps required to obtain the results, I would like to see the source code.

Moreover I would strongly suggest to make this public after publication (e.g., using GH).

## Major items

The RQ is: To what extend does peer-influence affect item adoption on GH.

My first comments relates to the usage of "item adoption".

Being nearly a decade old user of GH myself I have seen people use the "staring" system in a number of ways.

Moreover, the platform mechanisms, by which repos are suggested or highlighted to the user, has changed over time.

Regarding the star system:

- Some users on GH, use stars to *bookmark* interesting projects, and not to signal that they are or will use it

- Not all users who end up using the software in a repo star it.

- Not all repos are something to be used/adopted, but read (e.g., https://github.com/sindresorhus/awesome (116k stars at the time of writing))

- Some users, confuse the purpose of forks and stars and end up forking, when in reality they wanted to signal support.

What I want to say with this, is that equating "staring a repo" and "item usage" is not correct.

The work is still very valuable, however I would not use "item adoption" as the motivation, since the analogy to products is strenuous and as argued above "starring" is not the same as "usage".

The matching procedure is quite complex, and I had to reread it several times, and am still not sure I got all the steps, hence I would like to see the replication code.

An illustration (i.e., cartoon) of the steps would go a long way to clarify the procedure.

Why do you choose this particular period (February, 2011 to October, 2013) for the analysis? A sentence justifying this choice is necessary.

## Minor items

Regarding the role of "leaders":

Follower relations, like on twitter can be artificial. On GH, there are people with thousands of followers who follow just as many, therefore these follower relationships likely do not represent meaningful relations.

You use the term "preference types", without clear definition.

In line 48 you say that MS is interested in using the platform. You should note in the text that MS is the owner of GH as of 2019, or avoid the mentioning of MS.

- Fig.1(b) Not clear how to read this plot. It is a CCDF of something, not clear why CCDF and not PDF (histogram)

- Fig.1(d) something is missing: Where can I see the control group? There is only one curve in Figure~1~(d).

## Suggestion for future work

The following is just a suggestion, and in no way a critique of this work.

If the matching is done sufficiently in advance, the estimation could be turned into a Diff-in-Diff specification.

Here you would match treated and controls in a "recruitment phase" several months before the event, then look at "adoption" in the pre-treatment (after recruitment, before event) phase and post treatment phase.

Then you can compare pre-treatment and post-treatment "adoption" across treated and controls.

This is just a suggestion, and in no way a critique of this work.

Reviewer #2: The paper describes an interesting application of machine learning to causal inference for dynamic online networks, where numerous items diffuse simultaneously. Here, machine learning is used to infer agents' preferences (based on previously adopted decisions), which are subsequently used to match "treated" and "untreated" agents, and estimate causal effects. Here, "treated" means basically that one or more of the agents followed by the agent has recently adopted a specific item.

My major commet is that, in my opinion, the paper would improve a lot its readability (at least for a reader interested in technical details) by including in the main text some formulas (at least the most important ones): in this way, it would be easier for the reader to identify more quickly which are the variables of most interest in the analysis. If including other formulas were not possible (e.g., to make the paper more readable also to readers less interested in technical details), it would be nice at least to include a figure/diagram describing the basic steps of the proposed method, containing links to the parts of the paper where such steps are detailed.

The following are some other minor issues that the authors could consider to further improve the paper:

- sometimes, an empty space appears before a numbered reference (e.g., ( [2])). It would be better to remove it;

- by reading Section 1, one can conclude that "homophily" and "peer influence" are among the most important variables in the model. A link to their precise definitions (e.g., to the sections of the text/appendices where they are defined) would be very useful to correctly interpret the proposed model;

- the authors may wish to include some recent references about machine learning applied to causal inference, especially in the presence of heterogeneous causal effects;

- line 122: the term "relative peer influence" appears here for the first time: before, it was called "peer influence" (without "relative");

- line 14: "We will then include": i.e., in a second analysis?

- line 143: "Note that, unlike [3]": why "unlike" [3]? Here the authors should describe more clearly the difference between their model and the one contained in [3];

- line 161: "one possible approach would be to utilize the programming languages of past adoptions" -> this sentence could be improved by specifying that this is the case, e.g., of the GitHub application described later in the text;

- paragraph starting on line 201: in machine-learning terms, the "test set" mentioned here could be more properly called "validation set" (although some references in statistics still reserve the name "test set" to the case considered here). It is not clear, however, if "cross" validation os simply the hold out method has been performed;

- what is the interpretation of the off-diagonal elements of the matrix \\Sigma on line 213? Or has \\Sigma to be diagonal?

- in the paragraph starting on line 301, some more details about the shuffle test could be added;

- in the current version of the work, all the figures have been reported at the end of it, making it quite difficult to follow the part of the text where those figures are commented.

6. PLOS authors have the option to publish the peer review history of their article (what does this mean?). If published, this will include your full peer review and any attached files.

Reviewer #1: No

Reviewer #2: No

---

## [Author Response · Author response to Decision Letter 0]

27 Dec 2019

Reply to Referee 1

Thank you for your comments on our paper. We are please you liked the matching procedure and that you appreciate our robustness checks in conjunction with collaborative filtering. We believe the paper has greatly benefited from your constructive and thoughtful comments. 

In your letter, you highlight four main points that we should address for a successful revision: 1) share the source code, and make it public after publication, 2) improve the motivation, 3) provide an illustration of the matching algorithm, and 4) provide a justification for the choice of the time period covered by the data. We summarize these points and explain how we address each of these points in turn below. 

1. Source code. Because the procedure involves multiple steps you would like to better understand it by looking at the source code, and you strongly suggest to make this public after publication (e.g., using GH).

Our code is available at github.com/christopher-rojas/PeerEffectsGitHub, and the data at https://www.carojas.com/research. The code was there prior to our original submission, but we realized we did not write anywhere this information in the paper. Apologies for this oversight.

We have now cleaned the code further and we have uploaded the input data that can be used with our code to produce all of the output used in the paper. We now indicate in the paper where to find the data and code (see footnote 6, page 4).

2. Motivation. You suggested that we should be more careful in our choice of words when using "item adoption" as the motivation, since "starring" is not the same as "usage". 

Thank you for this comment. We viewed adoption as referring to starring behavior and that the research question was whether linked agents were starring the same behavior or not. We however see that without clarification this may be misleading. 

We now give in the introduction and in the materials section the definition of item adoption that we use in our empirical application (see lines 77-78 and 100-101) and in footnote 3 on page 3 we warn the reader that in this context it may not imply usage. In footnote 3 we also note that our method may have wider applicability. In addition to those clarifications, we have added in the concluding remarks (lines 637-640) notes of caution regarding the interpretation of our results due to the specific definition of item adoption in our empirical application. 

3. Illustration of the matching algorithm. Because our matching procedure involves multiple steps (see point 1 above), you suggest that an illustrative device (e.g. a cartoon) would be helpful to clarify the procedure.

Thank you again for this great suggestion. We have now added multiple figures to help clarify the procedure. In the Materials and Methods Section, we added a Process Map (Figure 1) which shows an example of what the Input and Output data “looks like,” as well as the steps in our empirical method required to produce the output. We have also added a graphical example to help illustrate our definition of treatment (Figure 2). 

4. Time window covered by data. You ask us why we chose February, 2011 to October, 2013 as a time window for our analysis. You note that a sentence justifying this choice is necessary.

February, 2011 is the earliest that data is available, and October, 2013 is the latest date for which we observe the link creation time. GitHub stopped providing this data after October, 2013.

We agree with you that we should have mentioned this argument. We now do so in the Data and descriptive evidence section in footnote 13 on page 9.

You also list a few minor comments. Your comments are in italics and are followed by our answers. 

Regarding the role of "leaders": Follower relations, like on twitter can be artificial. On GH, there are people with thousands of followers who follow just as many, therefore these follower relationships likely do not represent meaningful relations.

We do include all of the follower relations in our analysis. While we agree with you that not all follower relationship represent meaningful interactions, following an agent affects the information that agents receive. Any repo that the leader stars will lead to a notification in the activity feed of the follower. This notification is important in shaping behavior since our heterogeneity results suggest that exposure to information is the primary mechanism of peer influence.

On page 18, lines 561-587, we also specifically analyze heterogeneity for treatments in which the leader(s) had many followers, versus treatments in which the leader(s) had few followers. We do indeed find higher influence in the treatments by leader(s) with few followers. However, we also still find significant influence in the treatments by leader(s) with many followers.

You use the term "preference types", without clear definition.

We used the term preference types to indicate covariates which capture the types of repos starred in the past. These covariates could be the agent latent factors output by WRMF, or the fractions of adoptions in different languages, or binary indicators of whether or not an agent adopted a repo. However, we agree that the term preference types is unnecessary and can be confusing. So we have removed all references to preference types.

In line 48 you say that MS is interested in using the platform. You should note in the text that MS is the owner of GH as of 2019, or avoid the mentioning of MS.

We have deleted the reference since it is not an important fact for the paper. It has been replaced with the reference to an article which discusses the importance of open-source software for all of the big tech companies (lines 71-73).

- Fig.1(b) Not clear how to read this plot. It is a CCDF of something, not clear why CCDF and not PDF (histogram)

Figure 1(b) is now Figure 3(b), because of the figures we added describing our empirical method. In Figure 3(b) we plot the probability that an agent stars a repo, given that n of their leaders have already starred it, as n varies from 1 to 10. We have modified the sentence describing Figure 3(b) on page 9 (lines 343-347) to explicitly say that we graph the probability of adoption as a function of the number of adopting leaders.

- Fig.1(d) something is missing: Where can I see the control group? There is only one curve in Figure~1~(d).

Thank you for your careful reading. The graph, which is now Figure 3(d), has been fixed by adding the similarity for the control group----dyads of randomly matched agents.

You conclude your report with a suggestion for further research, which we hope is an example of how our work may inspire further research. You note that“If the matching is done sufficiently in advance, the estimation could be turned into a Diff-in-Diff specification. Here you would match treated and controls in a "recruitment phase" several months before the event, then look at "adoption" in the pre-treatment (after recruitment, before event) phase and post treatment phase. Then you can compare pre-treatment and post-treatment "adoption" across treated and controls.”

That’s a very good idea. We have not done this and hope to pursue it in a future paper.

 

Reply to reviewer 2

Thank you for your comments on our paper. We are glad that you found our application of machine learning interesting. We have incorporated all your suggestions in the revision. We believe the paper has greatly benefited from your feedback. 

In your letter, you provide a suggestion on how to improve the readability of the paper for a reader interested in technical details and a list of minor comments to further improve the paper.

We report your comments (in italics) and explain how we have addressed them in the paper below.

My major commet is that, in my opinion, the paper would improve a lot its readability (at least for a reader interested in technical details) by including in the main text some formulas (at least the most important ones): in this way, it would be easier for the reader to identify more quickly which are the variables of most interest in the analysis. If including other formulas were not possible (e.g., to make the paper more readable also to readers less interested in technical details), it would be nice at least to include a figure/diagram describing the basic steps of the proposed method, containing links to the parts of the paper where such steps are detailed.

We agree with you. As you surmise, however, because of the complexity of the algorithm there are too many even basic formulas that probably would not help the readability of the paper for a reader interested in the main ideas. Instead of including them we have added a Process Map which summarizes each of the main steps in our empirical method (Figure 1 on page 5), and provides snippets of hypothetical data. We have also added a visual example (Figure 2 on page 6) to better explain the details of our definition of treatment.

Minor issues 

- sometimes, an empty space appears before a numbered reference (e.g., ( [2])). It would be better to remove it;

We removed all the spaces.

- by reading Section 1, one can conclude that "homophily" and "peer influence" are among the most important variables in the model. A link to their precise definitions (e.g., to the sections of the text/appendices where they are defined) would be very useful to correctly interpret the proposed model;

We have added precise definitions for peer influence and homophily at the beginning of the Materials and Methods section. We define peer influence on page 3 (lines 104-107) as the direct causal effect of the leader’s starring on the starring outcome for the follower. We define homophily on page 4 (lines 124-129) as occurring when one agent follows another due to similarity in (observed or unobserved) attributes.

- the authors may wish to include some recent references about machine learning applied to causal inference, especially in the presence of heterogeneous causal effects;

Thank you for the suggested references. We have added footnote 1 on page 2 which lists recent references about machine learning applied to causal inference, including the application of machine learning to causal inference in the presence of heterogeneous causal effects.

- line 122: the term "relative peer influence" appears here for the first time: before, it was called "peer influence" (without "relative");

The word “relative” was used to indicate the fact that our method to estimate peer influence is based on a ratio, as opposed to a difference. However, we agree with you that the use of the term “relative peer influence” is redundant and confusing, so “relative” has been dropped. 

- line 140: "We will then include": i.e., in a second analysis?

What we mean is that the matching covariates may include both the baseline attributes, as well as the preference estimates from machine learning (or languages, binary adoptions). We have changed the sentence on page 4 (lines 136-138) to “In most of our empirical analyses, we also include in X_{u,i,t}additional covariates which summarize the types of items the agent adopted in the past to better proxy for preferences.”

- line 143: "Note that, unlike [3]": why "unlike" [3]? Here the authors should describe more clearly the difference between their model and the one contained in [3];

Thank you for this very helpful comment. “Unlike” should in fact be “like” since both [3] and our paper assume that social network link formation is independent of the adoption behavior. We have removed the line referenced above and we added a discussion of this common limitation to our concluding remarks on page 19 (lines 633-637). 

- line 161: "one possible approach would be to utilize the programming languages of past adoptions" -> this sentence could be improved by specifying that this is the case, e.g., of the GitHub application described later in the text;

Thanks again, on line 200 we replaced the line above with “we use the programming languages of items adopted in the past.”

- paragraph starting on line 201: in machine-learning terms, the "test set" mentioned here could be more properly called "validation set" (although some references in statistics still reserve the name "test set" to the case considered here). It is not clear, however, if "cross" validation os simply the hold out method has been performed

It would indeed be more accurate to refer to our hold-out set as a validation set rather than a test set. However, it is standard practice in many collaborative filtering academic papers (such as the WRMF paper by Hu et al, 2008) to refer to this set as simply a test set, because they don’t typically do cross-validation. In this revision, to solve any doubt we have followed your recommendation and replaced the term test set with validation set. See page 7 (line 243).

- what is the interpretation of the off-diagonal elements of the matrix \\Sigma on line 213? Or has \\Sigma to be diagonal?

The off diagonal elements are zero. We compute the variance-weighted Euclidean distance, as opposed to the Mahalanobis distance. We now explicitly noteon page 7 (line 256) that the matrix Sigma is a diagonal matrix.

- in the paragraph starting on line 301, some more details about the shuffle test could be added;

We have added more details about the shuffle test on page 10 (lines 350-359). We now say the following:

The original shuffle test is applied to a single item diffusing in an undirected social network. The test compares the clustering of adoption times between friends to the clustering between friends after permuting the adoption times such that the temporal frequency of adoptions remains constant. We implement the shuffle test in our setting by randomly reassigning the adoption times for each item (so that the adoption frequency for the item over time remains constant), and we compare the distribution of observed dyadic (leader-follower pair) differences in adoption times for the original data relative to the data with the shuffled adoption times. Since our network is directed, we limit our comparison to observations in which the leader adopts before the follower. 

- in the current version of the work, all the figures have been reported at the end of it, making it quite difficult to follow the part of the text where those figures are commented.

The figures have been added to the text.

---

## [Editor Report · Decision Letter 1]

15 Jan 2020

Multidimensional diffusion processes in dynamic online networks

PONE-D-19-23200R1

Dear Dr. Easley,

We are pleased to inform you that your manuscript has been judged scientifically suitable for publication and will be formally accepted for publication once it complies with all outstanding technical requirements.

With kind regards,

Massimo Riccaboni

Academic Editor

PLOS ONE
---

## [Editor Report · Acceptance letter]

23 Jan 2020

PONE-D-19-23200R1 

Multidimensional diffusion processes in dynamic online networks 

Dear Dr. Easley:

I am pleased to inform you that your manuscript has been deemed suitable for publication in PLOS ONE. Congratulations! Your manuscript is now with our production department. 

With kind regards,

on behalf of

Dr. Massimo Riccaboni 

Academic Editor

PLOS ONE